# Alteration of the Nucleotide Excision Repair (NER) Pathway in Soft Tissue Sarcoma

**DOI:** 10.3390/ijms23158360

**Published:** 2022-07-28

**Authors:** Adriano Pasqui, Anna Boddi, Domenico Andrea Campanacci, Guido Scoccianti, Andrea Bernini, Daniela Grasso, Elisabetta Gambale, Federico Scolari, Ilaria Palchetti, Annarita Palomba, Sara Fancelli, Enrico Caliman, Lorenzo Antonuzzo, Serena Pillozzi

**Affiliations:** 1Medical Oncology Unit, Careggi University Hospital, 50134 Florence, Italy; apasqui.oncclinica@gmail.com (A.P.); lorenzo.antonuzzo@unifi.it (L.A.); serena.pillozzi@unifi.it (S.P.); 2Orthopaedic Oncology Unit, Careggi University Hospital, 50134 Florence, Italy; anna.boddi@gmail.com (A.B.); domenicoandrea.campanacci@unifi.it (D.A.C.); scocciantig@aou-careggi.toscana.it (G.S.); scolari.fed@gmail.com (F.S.); 3Orthopaedic Oncology Unit, Careggi University Hospital, Department of Health Sciences, University of Florence, 50134 Florence, Italy; 4Department of Biotechnology, Chemistry and Pharmacy, University of Siena, 53100 Siena, Italy; daniela.grasso@student.unisi.it; 5Clinical Oncology Unit, Careggi University Hospital, 50134 Florence, Italy; gambalee@aou-careggi.toscana.it; 6Department of Chemistry Ugo Schiff, University of Florence, 50019 Sesto Fiorentino, Italy; ilaria.palchetti@unifi.it; 7Histopathology and Molecular Diagnostic Unit, Careggi University Hospital, 50134 Florence, Italy; annarita.palomba@aouc.unifi.it; 8Department of Experimental and Clinical Medicine, University of Florence, 50134 Florence, Italy; sara.fancelli@unifi.it (S.F.); enrico.caliman@unifi.it (E.C.)

**Keywords:** nucleotide excision repair (NER), soft tissue sarcoma, ERCC, SNP, pharmacogenomic

## Abstract

Clinical responses to anticancer therapies in advanced soft tissue sarcoma (STS) are unluckily restricted to a small subgroup of patients. Much of the inter-individual variability in treatment efficacy is as result of polymorphisms in genes encoding proteins involved in drug pharmacokinetics and pharmacodynamics. The nucleotide excision repair (NER) system is the main defense mechanism for repairing DNA damage caused by carcinogens and chemotherapy drugs. Single nucleotide polymorphisms (SNPs) of NER pathway key genes, altering mRNA expression or protein activity, can be significantly associated with response to chemotherapy, toxicities, tumor relapse or risk of developing cancer. In the present study, in a cohort of STS patients, we performed DNA extraction and genotyping by SNP assay, RNA extraction and quantitative real-time reverse transcription PCR (qPCR), a molecular dynamics simulation in order to characterize the NER pathway in STS. We observed a severe deregulation of the NER pathway and we describe for the first time the effect of SNP rs1047768 in the *ERCC5* structure, suggesting a role in modulating single-stranded DNA (ssDNA) binding. Our results evidenced, for the first time, the correlation between a specific genotype profile of ERCC genes and proficiency of the NER pathway in STS.

## 1. Introduction

Soft tissue sarcomas (STS) are a heterogeneous group of rare malignancies with mesenchymal origin with an incidence of approximately 50–60 new cases per year for every million people, 20% with synchronous metastasis at diagnosis [1,2]. Approximately 30–40% of localized STS will finally develop metastasis. Generally, surgery with or without radiation, even if only in selected cases of localized disease, represents the ideal treatment for STS [3]. For locally advanced or metastatic STS, the therapeutic options are limited and clinical responses to chemotherapy (ChT) are uncommon, while the median of survival ranges between 18 and 22 months [3]. Anthracycline-based therapy is the backbone of first-line therapy for advanced disease and there are no formal data that show that the combination of ChT regimens is related to an improved overall survival (OS) than single-agent ChT with doxorubicin [4,5,6]. A higher response rate and longer progression-free survival (PFS) can be expected in a number of selected for probable better sensitivity histological subtypes in light of evidence from several randomized, clinical trials [7,8,9]. However, clinical responses to anticancer therapies continue to be limited to a small subset of STS patients. Thus, the detection of biomarkers that can predict drug response and toxicity can be useful to explain the basis for the differences in treatment efficacy and toxicity among patients.

In the last decade, pharmacogenetic and pharmacogenomic approaches have been increasingly applied to the genetic characterization of STS and have started to offer insights that may be considered to modulate and, hopefully, improve the effectiveness and safety of the currently used treatment regimens [10].

Most cancer therapy regimens damage DNA, killing cancer cells by inducing damage that interferes with DNA replication and transcription, finally activating cell death pathways. The efficient repair of DNA damage can render cells resistant to killing. The DNA repair pathways involved in acquired resistance mechanisms vary from drug to drug, and all of the major DNA repair pathways can alter the sensitivity of cancer cells to DNA damaging agents and thus confer chemoresistance. It has been suggested that deficiencies in DNA repair capacity could have a role in affecting the response to DNA damaging agents [11,12,13]. At least six major DNA repair pathways, as well as numerous sub-pathways, are involved in the repair of damage [11,14]. In particular, the nucleotide excision repair (NER) pathway (Figure 1) repairs bulky lesions and has been associated with tumor progression and response to chemotherapy [15,16]. Excision repair cross-complementing (ERCC) group 1 (*ERCC1*/XPA), group 2 (*ERCC2*/XPD), group 4 (*ERCC4*/XPF) and group 5 (*ERCC5*/XPG) are members of the NER pathway. It has been demonstrated that a deficiency of DNA repair capacity of NER genes can be related to the presence of single nucleotide polymorphisms (SNPs) that result in altered mRNA expression or protein activity [14]. A deficiency in the ability to repair DNA damages could therefore alter the response to treatment with a significant positive association between polymorphism *ERCC2* gene, Lys751Gln, but also *ERCC1*, Asn118Asn with the improved cisplatin response in osteosarcoma [17]. Moreover, emerging evidence associates the expression levels of genes involved in the NER process and some types of cancer. A study of 55 patients and a larger study of 744 patients with squamous cell carcinoma of head and neck revealed that patients had a reduced DNA repair capacity, reduced expression of *DDB1* and *ERCC3* genes, compared to healthy controls, resulting in an increased risk of developing cancer [18,19]. In patients with colorectal cancer *ERCC1*, *ERCC2*, *ERCC5* and *DDB2* genes are more expressed in tumor tissue than in matched normal tissue, while *ERCC4* is downregulated [20]. A bioinformatics analysis performed on the GSE28460 and GSE18497 expression datasets showed that, in pediatric acute lymphoblastic leukemia patients, NER pathway gene expression is significantly increased at relapse; moreover, at diagnosis, NER pathway gene expression is higher in early relapsing subpopulation than in the late relapsing group [21].

In the present study, in order to identify and characterize the role of the NER pathway in STS, we evaluate the expression of key genes (*ERCC1*, *ERCC2* and *ERCC5*) and a panel of SNPs within these genes in a cohort of STS patients. Finally, with a molecular dynamics simulation, we predicted the possible effect of selected SNPs on the secondary and tertiary structure of NER proteins.

## 2. Materials and Methods

### 2.1. STS Patients and Samples

A total of 32 STS tissues, swab and plasma samples were obtained from patients enrolled at the Oncology and Reconstructive Orthopedics SOD, AOU Careggi (RESEARCH study) after written consent. The main clinical and pathological characteristics of the patients are summarized in Table 1. STS tissue samples were used for total DNA and RNA extraction. Swab samples were used for genomic DNA extraction.

### 2.2. Genomic and Somatic DNA Extraction and Genotyping by SNP Assay

Genomic and somatic DNA was isolated from buccal swabs and tumor tissues using a QIAamp DNA Blood mini kit (Qiagen, Manchester, UK). Allelic discrimination was performed using TaqMan SNP Genotyping Assays (Applied Biosystems, Forster City, CA, USA) on the Rotor-gene 6000 instrument (Qiagen, Manchester, UK). The assay consisted of two allele-specific minor groove binding (MGB) probes, labeled with the fluorescent dyes VIC and FAM.

Real-time PCR was performed in 15 μL reaction mixtures containing 7.5 μL of Taqpath ProAmp MasterMix (Applied Biosystems, Forster City, CA, USA). TaqMan-MGB genotyping assay mixes were supplied at 40× concentration and 2 μL of sample DNA. Thermocycler conditions were an initial hold step for 30 s at 60 °C and 5 min at 95 °C, followed by 40 cycles of 95 °C for 15 s and 60 °C for 60 s, finally a last post read step for 30 s at 60 °C. Genotypes were analyzed by measuring allele-specific fluorescence using the Rotor-gene Q Series software 2.3.5 for allelic discrimination (Qiagen, Manchester, UK).

### 2.3. RNA Extraction and Quantitative Real-Time Reverse Transcription PCR (qPCR)

Total RNA was extracted from STS tissue samples using the RNeasy Mini kit (Qiagen, Manchester, UK). The quantity and the quality of RNA were evaluated using a Nanodrop spectrophotometer version 3.5 (NanoDrop Technologies, Wilmington, DE, USA). A total of 50 ng of RNA from each sample was retro-transcribed with the PrimeScript™ RT Reagent Kit (Takara Bio, Otsu, Shiga, Japan); the resulting cDNA was then used for qPCR analysis with “Powerup Sybr Green” (Applied Biosystems, Forster City, CA, USA) on a Rotor Gene Q (Qiagen, Manchester, UK). The relative quantification was performed using LinRegPCR software (version 11.0, http:/LinRegPCR.nl) and the data were normalized to 18s rRNA. The primers used are listed in Table 2.

### 2.4. Molecular Dynamics Simulations

Molecular dynamics simulations were carried out in the AMBER99sb force field employing the GROMACS version 2022.2 [22] software suite. For ERCC2, the cryo-EM structures 6NMI and 5OF4 were employed. The PDB entry 6TUX was taken as starting structure for the ERCC5-DNA complex; the model of activated ERCC5-DNA complex was built using the DNA in the T4 RNase H crystal structure (PDB 2IHN). Mutants were generated by PyMOL (Schrödinger Inc., New York, NY, USA) mutagenesis subroutine. Each structure was placed in a triclinic box of dimensions exceeding protein size by 1 nm each side after alignment along inertia axes; boxes were filled with TIP3P water molecules and counterions were added until neutrality. The simulation protocol involved an initial restrained minimization step, followed by a series of equilibration MD simulations in the NVT ensemble, during which temperature was gradually increased from 0 to 300 K, and restraints on protein heavy atoms were gradually revealed. Furthermore, unrestrained equilibration in the NPT ensemble was performed during 500 ps, prior to a 1000 ns production stage. The equations of motion were integrated with a time step of 2.0 fs. Periodic boundary conditions were used, in conjunction with the particle-mesh Ewald (PME) method. All X-H bonds were constrained using the LINCS algorithm. Snapshots were dumped to a trajectory file each picosecond. The trajectory was submitted for further analysis after the routine evaluation of correctness (i.e., RMSD and RMSF).

### 2.5. Statistical Analysis 

Statistical analysis was performed using RStudio on relative expression values obtained from qPCR, genotyping data derived from real time PCR and clinical pathological characteristics of enrolled patients. χ2 tests were performed to compare the distribution of selected SNPs between our patient cohort and the general population. Haplotype-base evaluation of SNPs in *ERCC1*, *ERCC2* and *ERCC5* genes was performed using the HaploView software version 4.2 (http://www.broad.mit.edu/haploview). The hierarchical clustering heatmaps were created with the pheatmap package (version 1.0.12) in RStudio using Euclidean correlation, while the correlation matrix was created with the GGally package (version 2.1.2) using Pearson’s correlation. Levene’s test significance on the dataset disproved the assumption of equality of variance; therefore, the relative expression values of the three *ERCC* genes in every patient was analyzed with Welch’s *t*-test.

## 3. Results

### 3.1. Clinicopathological Features of OS Patients

The clinicopathological features of the 32 patients affected by STS (62.5% male and 37.5% female) enrolled in the study are illustrated/shown in Table 1. The median age of the patients is 68.4 years (range of 39–93), 20 patients (62.5%) had tumor localized in the thigh, 1 in foot (3.1%), 2 in gluteal-sacral region (6.25 %), 1 in scapular region (3.1%), 1 in deltoid region (3.1%), 1 in abdominal wall (3.1%), 2 in pectoral region (6.25%) and 4 in upper limb (12.5%). Regarding the histotype, 9 (28.1%) patients had liposarcoma (4 pleomorphic and 5 myxoid), 12 patients (37.5%) had pleomorphic sarcoma, 3 patients leiomyosarcoma (9.4%), 4 patients myxofibrosarcoma (12.5%), 1 patient fibrosarcoma (3.1%), 1 fibromyxoid sarcoma (3.1%) and 1 dermatofibrosarcoma protuberans (3.1%), while 1 was undifferentiated pleomorphic cell sarcoma (3.1%); in addition, 25 (78%) patients were affected by high-grade STS and 7 patients were affected by low-grade STS (22%); 6 (19%) patients relapsed. The median value of Body Mass Index (BMI) was 25.5 kg/m^2^ (range 19.2–44.3). Finally, large surgical margins were registered in 90.6% of cases.

### 3.2. Evaluation of Germinal SNP Panel in NER Genes 

Given the abundant evidence on the correlation between polymorphisms in NER genes and different types of cancer, we decided to evaluate the possible role of SNPs in the NER pathway in STS patients. In our cohort, we analyzed, at the germline level, a panel of SNPs belonging to relevant genes of the NER pathway: the synonymous variant rs11615 (chr19:45420395) of *ERCC1*; the stop gained variant rs13181 (chr19:45351661) and the missense variant rs1799793 (chr19:45364001) for *ERCC2*; and the missense variant rs1047768 (chr13:102852167) and the upstream variant rs2296147 (chr13:102846025) for *ERCC5* (Table 3). One patient was excluded from the analysis as the amount of DNA extracted from his swab sample was not sufficient. We then compared the allele frequencies of our cohort to the healthy population (https://www.ncbi.nlm.nih.gov/snp/, accessed on 2 March 2022). The germline allele and genotype frequencies of the selected SNPs registered in our cohort of STS patients are summarized in Table 3. Regarding the SNP rs11615 (A>G) in the *ERCC1* gene, the homozygous reference condition (AA) is present in 19%, the heterozygous condition (AG) is present in 58% and the alternative homozygous condition (GG) in 23% of STS patients (A = 48 %; G = 52 %); the alternative allele has a higher frequency than the general population. Regarding the SNP rs13181 in the *ERCC2* gene (T>G; TT = 19%, TG = 45%, GG = 36%; T = 42%, G = 58%), the alternative allele has a higher frequency than the general population. Regarding the SNP rs1799793 in the *ERCC2* gene (C>T; CC = 19%, CT = 55%, TT = 26%; C = 47%, T = 53%), the alternative allele has a higher frequency than the general population. For the SNP rs1047768 in the *ERCC5* gene (T>C; TT = 19%, TC = 45%, CC = 36%; T = 42%, C = 58%), the alternative allele has the same frequency as the general population. Finally, regarding the SNP rs2296147 in the *ERCC5* gene (T>C; TT = 32%, TC = 49%, CC = 19%; T = 56%, C = 44%), the alternative allele has a lower frequency than the general population. For both SNPs in the *ERCC2* gene, this difference was statistically significant (Table 3) (rs13181 χ^2^ = 9.72 *p* = 0.0018; rs1799793 χ^2^ = 9.02 *p* = 0.0027). The presence or absence of the studied variants are summarized in Figure 2, which shows that the rs11615 variant in the *ERCC1* gene is present in 25 out of 31 patients, the rs13181 and rs1799793 variants in the *ERCC2* gene are present in 25 patients and finally the rs1047768 and rs2296147 variants in the *ERCC5* gene are present in 25 and 20 patients, respectively. In our patient cohort, 11 (35.5%) patients had all SNPs analyzed, while only 1 patient had no variations.

### 3.3. Linkage Disequilibrium and Haplotypes

We calculated the linkage disequilibrium (LD) coefficients (D’ and r2) to evaluate the non-random association of alleles at different loci, and frequency of haplotypes using Haploview version 4.2 (Mark Daly’s Laboratory, Massachusetts Institute of Technology/Harvard Broad Institute, Cambridge, MA, USA) in *ERCC* genes. As shown in Figure 3A, the LD value between the variants of *ERCC5* is 54, indicating a low value of inkage disequilibrium (white block) and the haplotype with the highest frequency is CC. For the variants concerning *ERCC1* and *ERCC2*, as shown in Figure 3B, we found that the *ERCC2* variants rs13181 and rs1799793 show a higher LD value, 69 (red block), instead of *ERCC1* variants. The haplotype with the highest frequency for *ERCC1*-*ERCC2* variants is TGG. 

### 3.4. Evaluation of Somatic SNP Panel in NER Genes in STS Patients

In addition to germline SNPs, somatic variations in *ERCC* genes have also been shown to play a role in different tumor types, such as non-small cell lung cancer [23] and urothelial carcinoma [24]. In the cohort of patients described above, we investigated whether the presence of a wild-type genotype recurs also in the somatic line, so we analyzed the same variants at the somatic level. For *ERCC1* rs11615 and *ERCC2* rs13181 and rs1799793, the homozygous reference condition was also maintained at the somatic level in all patients, whereas in *ERCC5* rs2296147 and rs1047768, we found that in a 12,5% and 20%, respectively, of homozygous reference patients there was a change in genotype to the mutated form. Frequencies revealed in the somatic line are summarized in Table 4. 

### 3.5. Expression Analysis of NER Genes in STS Patients by qPCR

Emerging evidence associates the expression levels of genes involved in the NER process and some types of cancer, while data on STS are limited, so we evaluated the gene expression of three key genes (*ERCC1*, *ERCC2* and *ERCC5*) in the NER pathway in 24 tissue samples from the 32 patients in our cohort. As illustrated in Figure 4, the gene *ERCC1* is overexpressed in 75% (18/24) of STS samples analyzed compared to healthy corresponding tissue with a median expression value of 0.352 (range of −2.32; 2.68). The *ERCC2* gene is overexpressed in 50% (12/24) STS with a median expression value of 0.497 (range −7.04; 3.04). Finally, the *ERCC5* gene is overexpressed in 42% (10/24) STS samples analyzed with a median expression value of −0.607 (range of −4.66; 2.22). 

We then investigated if a distinct genotype combination of *ERCC1*, *ERCC2* and *ERCC5* genes could influence the expression levels of these genes in the tumor tissue. 

We divided the samples by whether they presented the wild-type allele in a SNP and compared the expression level of the resulting two groups (Figure 5). We found that patients with AA or AG genotypes in the rs11615 SNP have lower expression of the *ERCC1* gene than mutated homozygous GG (A); patients with TT or TG genotypes in the rs13181 SNP have lower expression of the *ERCC2* gene than mutated homozygous GG (B); patients with CC or CT genotypes in the rs1799793 SNP have significantly lower expression of the *ERCC2* gene than patients with both mutated alleles TT (C, *p* = 0.0316); and finally, patients with TT or CT genotype in the rs2296147 and rs1047768 have a slightly higher expression of the *ERCC5* gene than mutated homozygous CC (D,E). The expression values of *ERCC1*, *ERCC2* and *ERCC5* were compared with each other using R, and the results were represented as a heatmap as Pearson’s correlation values (Figure 6B), showing the high level of correlation between the expression of *ERCC2* and *ERCC5*.

We compared the expression level of each gene in each sample using the R package pheatmap. We observed that the expression level does not generate a cluster of samples in their respective histotype, sex or grade (Figure 6A).

A statistical analysis was then performed to see how the characteristics of the patients are correlated to the expression levels of the NER genes, and we observed that, in our cohort, female patients had significantly higher expression levels of *ERCC1* than male patients (*p* = 0.02).

### 3.6. Correlation between TP53 and NER Genes

It has been reported that allelic variation at SNPs on the *TP53* gene affects *ERCC5* transcript and this could result in variation in NER function in normal bronchial epithelial cells [25]; in particular, T allele at SNP rs2296147 upregulates and C allele at SNP rs17655 downregulates *ERCC5* expression levels. The common SNP sites rs751402 and rs2296147 reside in the *ERCC5* 5′-untranslated region (UTR). Variation at rs2296147 is predicted to alter the binding of the *TP53* transcription factor [26]. Notably, rs2296147 T allele plays part in the formation of an in silico predicted *TP53* transcription factor-binding site and that site is predicted to be lost when the C allele is present. Based on these premises, we evaluated the *TP53* gene for the SNP rs1042522 a missense variant (G>C) in our cohort of STS patients; the homozygous reference condition was present in 5%, the heterozygous condition was present in 50% and the alternative homozygous condition in 45% of STS patients. Then, we evaluated the correlation between *TP53* genotyping and expression level of *ERCC5* and other *ERCC* genes, finding no relevant differences. 

### 3.7. Effect of SNPs on the Tertiary Structure of NER Proteins

The SNPs rs13181 and rs1799793 (*ERCC2*) and rs1047768 (*ERCC5*) were evaluated for their effect on protein structure folding and stability, and also in view of linkage disequilibrium. In ERCC2, Asp312 is part of a markedly negatively charged loop spanning residues Pro311-Pro320 (Figure 7A), with three carboxylic sidechains driving the conformation thanks to the apparent ionic pairs: Asp312-Lys412, Glu313-Arg373 and Glu317-Lys370. Moreover, more cross-interactions between opposite charges are established within the 12 Å limit of the Coulomb cutoff threshold, forming a surface electrostatic interaction network stabilising the local structure. Indeed, the molecular dynamics simulation of the native ERCC2 showed an unusual stability level for such a loop, comparable to that of a secondary structure (Figure 8). Consequently, the simulation of the Asp312Asn mutant (rs1799793) shows such segment to be destabilized by the loss of negative charge and the consequent disruption of the electrostatic network. The asparagine residue cannot offer a replacement interaction such as, e.g., hydrogen bonding because the distance to neighbouring sidechains exceeds even the largely allowed 3.4 acceptor–donor distance. The other variant of ERCC2 (rs13181) is of the stop gain type and causes the loss of the C-terminal segment (Figure 7B) starting at Lys751; half of the α-helix and the final β-strand Ala756-Q758 and relative pairing with Ala479-Phe481 are lost. Molecular dynamics simulation showed such loss of secondary structures does not disrupt the overall folding, but affects local stability significantly (Figure 7B). Although both SNPs seem not to disrupt the overall folding, a cooperative effect in lowering the stability of tertiary structure, with a possible cascade effect on Transcription factor IIH (TFIIH) complex formation, can be suggested to explain the linkage disequilibrium effect.

The aminoacidic position His46 of ERCC5 is central to the hydrophobic wedge (residues 31–67), including helix α2 and helix α3 and their connecting loop, which is mostly disordered. A positively charged canyon defined by the hydrophobic wedge and β-pin motifs was proposed to provide a DNA-binding surface to the well-known helical arch formed by α-helices α4 (residues 82–97) and α6 (residues 750–779) [27] and H2TH motif (residues 848–880) [28]. To test the possible role of wedge His46 in binding DNA, ERCC5 in complex with the 3′ side of the repair bubble was modelled, using the structure of the FEN1 homolog T4 RNase H, where the path of both ssDNA strands beyond the junction was observed [29]. The model was refined by molecular dynamics simulation, resulting in a clash-free structure showing both DNA strands fit in protein crevices. The undamaged ssDNA fits in the crevice formed by the hydrophobic wedge and the β-pin motifs. Such crevice was already reported to accommodate the phosphate backbone by interaction with basic amino acids H60 and R61 in the wedge helix α3, K828 in the β-pin, and K972 in α15. Our model showed the His46 to contribute to the binding with a similar interaction but with the solvent-exposed phosphate of the first decoupled nucleobase, stabilizing the interaction at ss/dsDNA junction level (Figure 9). The model of the His46Gln mutant showed a shift of the wedge dynamics back to the disordered state possibly affecting the wedge effectiveness in opening the double strand.

## 4. Discussion

The scientific community, still aims to improve and optimize the management of STSs, because of their heterogeneity and rarity. The pathogenetic and histopathological knowledge of STS has increased in recent decades, but unfortunately without the desired impact on patient survival rates. However, a higher response rate and longer PFS can be expected in a number of selected for probable better sensitivity histological subtypes in light of evidence from several randomised clinical trials. For example, doxorubicin plus dacarbazine is an optional multi-agent first-line regimen for leiomyosarcoma, in which the activity of ifosfamide is debated [6]. Gemcitabine represents a therapeutic option, alone or in combination with docetaxel [30]. Notably, a phase III study compared single-agent doxorubicin with the combination of gemcitabine–docetaxel as first-line regimen in advanced STS patients of all types. The combination did not show improvement in PFS and objective response rate (ORR) and is not recommended as upfront therapy for advanced STS patients [8], including uterine LMS. Taxanes can be an effective treatment options for angiosarcoma, which has shown high sensitivity to this ChT [9]. Neurotrophic tyrosine receptor kinase (NTRK) inhibitor larotrectinib represent the standard treatment of those rare patients with locally advanced or metastatic NTRK rearranged sarcomas [31]. Second-line treatments include pazopanib and chemotherapeutic agents such as trabectedin, eribulin, and gemcitabine combinations. The median PFS in first-line therapy for STS ranges between 4.5 and 6.0 months, whereas the median PFS for most of the second-line drugs is below 5 months [32,33,34]. Therefore, there is an urgent need to increase the knowledge behind these diseases and to develop more effective diagnostic and prognostic strategies.

The analysis of data in the literature and the importance of DNA repair mechanisms in tumor progression led us to the characterization of the NER pathway, the DNA repair pathway often implicated in chemoresistance processes [35,36,37], in a cohort of STS patients enrolled at the Orthopaedic Oncology Unit of Careggi University Hospital. It has been demonstrated that efficiency changes of DNA repair capacity of NER genes can be related to the presence of single nucleotide polymorphisms (SNPs) that result in altered mRNA expression or protein activity [14]. A deficiency in the ability to repair DNA damages could therefore alter the response to treatment with a significant positive association between the polymorphism *ERCC2* gene, Lys751Gln, but also *ERCC1*, Asn118Asn with the improved cisplatin response in osteosarcoma [17]. Additionally, variant alleles in NER SNPs *ERCC2* Lys751Gln, *ERCC1* 8092 C/A and *ERCC1* 118C/T were individually associated with esophageal adenocarcinoma risk [38]. *ERCC2* rs50871 G/T, *ERCC6* rs1917799 G/T and *DDB2* rs3781619 A/G polymorphisms were significantly associated with shorter OS, while *ERCC1* rs3212961 A/C, *ERCC5* rs2094258 A/G and *DDB2* rs830083 C/G could predict favorable OS of gastric cancer patients [39]. The *ERCC2* rs3810366 and rs238406 polymorphisms were shown to significantly enhance Wilms tumor susceptibility [40]. There is also evidence in the literature on the association between SNPs in NER genes and STS, e.g., *ERCC5* and *ERCC1* status represented a potential DNA repair signature that could be used for the prediction of clinical response to trabectedin in STS [41] and to irofulven in urothelial cancer [42]. In addition, the genotyping of the SNPs rs17655 (Lys751Gln) in *ERCC5* and rs13181 (Asp1104His) in *ERCC2* revealed the potential implication of the Asp1104His polymorphism in the occurrence of chromosomal translocations associated with specific sarcoma subtypes [43].

These data led us to evaluate a panel of SNPs in germline *ERCC* genes in a cohort of STS, considering that the alternative allele for SNP rs11615 in the *ERCC1* gene and the alternative allele for SNPs (rs13181and rs1799793) in the *ERCC2* gene has a higher frequency in STS patients versus the distribution in the general healthy population and NER pathway is deeply affected since 97% (30 out of 31) of STS patients have at least one variation in *ERCC* genes. In addition, rs1799793 and rs13181 variants on *ERCC2* shown a higher linkage value, as also reported in previous studies conducted with different populations [44,45].

We then evaluated the presence of somatic mutations on *ERCC* genes in those patients presenting a homozygous reference genotype, observing that 20% (rs1047768) and 12.5% (rs2296147) of patients have somatic variant on *ERCC5*; therefore, both germline and somatic line are involved in our cohort of patients. 

The altered expression of *ERCC* genes has been associated with increased risk of colorectal cancer [46], where *ERCC1*, *ERCC2* and *ERCC5* were overexpressed, and invasive cervical carcinoma [47], where *ERCC1* and *ERCC2* were underexpressed. In addition, the higher expression levels of the *ERCC1* gene have been linked with poor response of FOLFOX-based chemoradiation in colorectal cancer [48] and with the aggressiveness of esophageal cancer [49]. Furthermore, *ERCC1* expression has been correlated with trabectedin sensitivity in STS patients [50]. Data available in the literature on this type of rare tumor are extremely limited, and the most relevant dataset to date is TCGA-SARC (Appendix A) [51]. In this study, it was found that *ERCC1*, *ERCC2* and *ERCC5* were expressed in all of the STS tissue samples selected within our cohort. Subsequent genotyping of *ERCC* genes was performed to determine how differences at the germline level could be associated with altered expression levels and the results show that the *ERCC2* gene has significatively lower expression values in patients presenting the wild-type C allele in the rs1799793 variant. 

A subset of the observed SNPs (namely, rs13181, rs1799793 and rs1047768) on NER pathway genes, being of the missense or stop gained type, were suitable for the structural modeling of the induced three-dimensional structures alteration of NER proteins. The correlation with protein folding stability, also in view of linkage disequilibrium, is here discussed. The SNPs rs13181 and rs1799793 code for Asp312Asn and Lys751* mutants, located in the Arch domain and in the C-terminal CTE domain, respectively. The Arch domain is known to be mechanistically important for transcription and DNA repair via its binding to MAT1 [52]; dynamics alteration induced by Asp312Asn affects the loop closely interacting with MAT1 (Figure 7). The CTE domain has been suggested to form an interaction surface with the p44 protein [53], and modeling of Lys751* mutant showed a large disruption of such surface. Taking everything into consideration, Asp312Asn and Lys751* modeled as double mutant showed no effect on overall protein folding stability, but the co-operative local alterations of shape and dynamics affecting the interaction network also explain linkage disequilibrium. Coming to ERCC5, we reported how His46 contributes to the binding of ssDNA with the solvent-exposed phosphate of the first decoupled nucleobase, stabilizing the interaction at ss/dsDNA junction level (Figure 9). Molecular dynamics simulation the His46Gln mutation (rs1047768) showed a shift of the wedge dynamics back to the disordered state possibly affecting the wedge effectiveness in opening the double strand. Although the critical role of many basic residues in the wedge area has been described [26], to our knowledge, this is the first structural rationale proposed for His46 mutation.

In conclusion, we registered a severe deregulation of NER pathway in STS patients, with alterations in genotyping and expression of *ERCC* genes, i.e., the correlation between *ERCC2* expression and its genotype. Finally, the effect of SNP rs1047768 in the ERCC5 structure was hypothesized for the first time in this study suggesting a role in modulating ssDNA binding, a result that needs to be further validated, improving knowledge and optimizing the management of STSs.

## Figures and Tables

**Figure 1 ijms-23-08360-f001:**
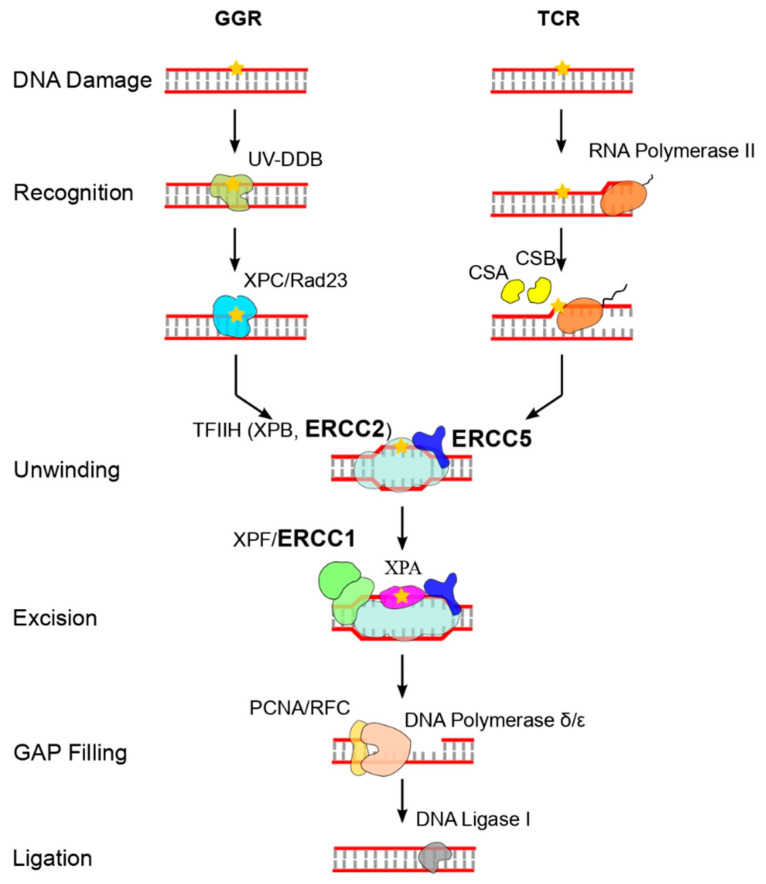
Nucleotide excision repair pathway (NER). GGR, global genomic NER; TCR, transcription coupled NER.

**Figure 2 ijms-23-08360-f002:**
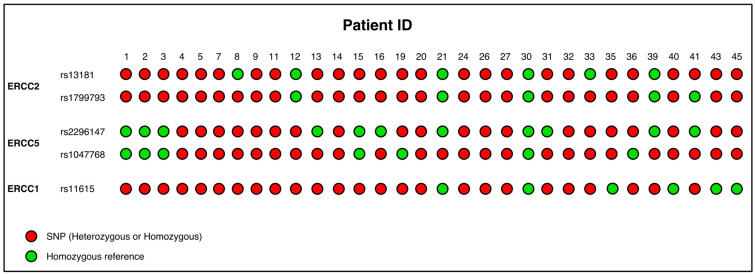
Presence/absence of the ERCC variants in the STS cohort. STS, soft tissue sarcoma.

**Figure 3 ijms-23-08360-f003:**
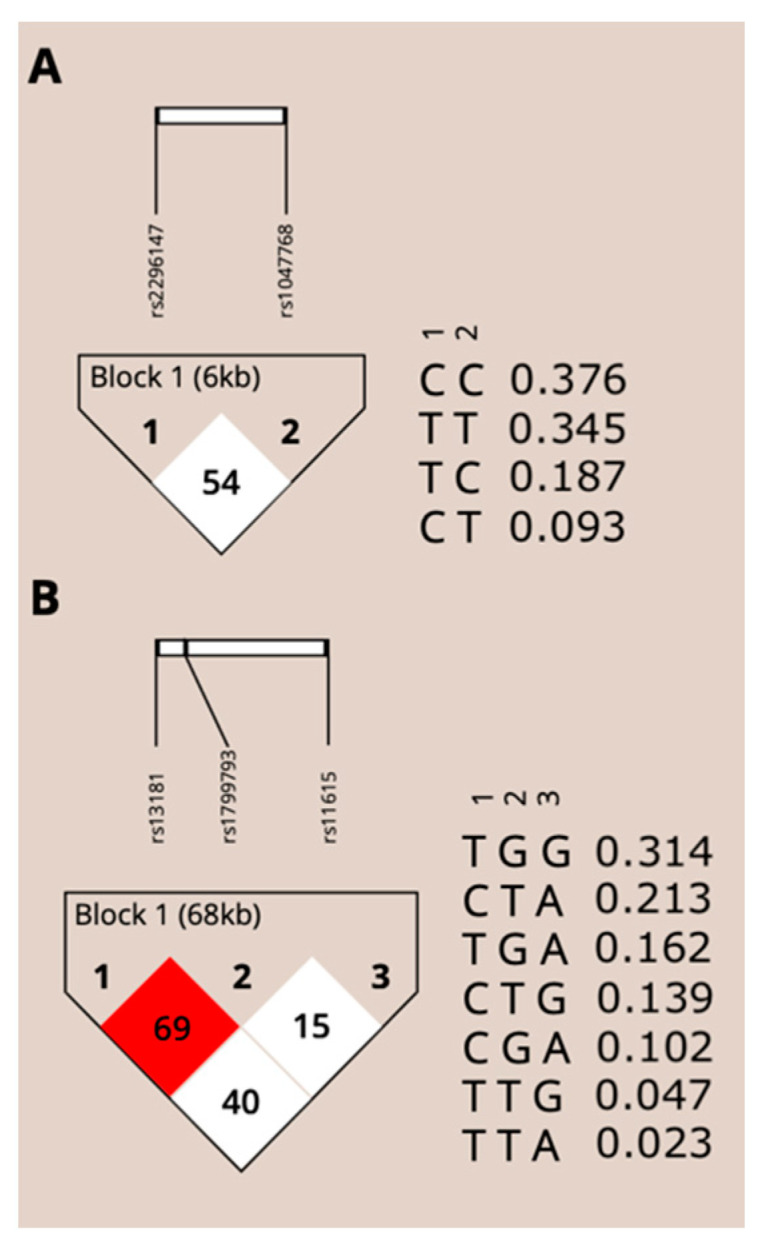
(**A**) Linkage disequilibrium (LD) block and haplotype frequencies for rs2296147 and rs1047768 in ERCC5 gene. (**B**) LD block and haplotype frequencies for r1318, rs799793 and rs11615 in ERCC1 and ERCC2 genes.

**Figure 4 ijms-23-08360-f004:**
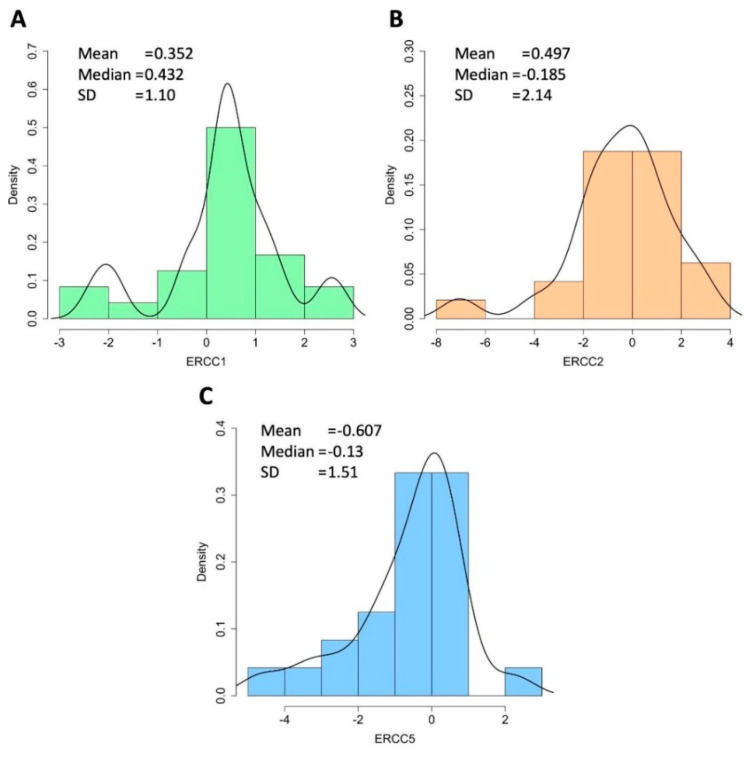
Expression plots showing trends of individual NER genes in STS samples. (**A**) ERCC1; (**B**) ERCC2; (**C**) ERCC5. SD, standard deviation; STS, soft tissue sarcoma.

**Figure 5 ijms-23-08360-f005:**
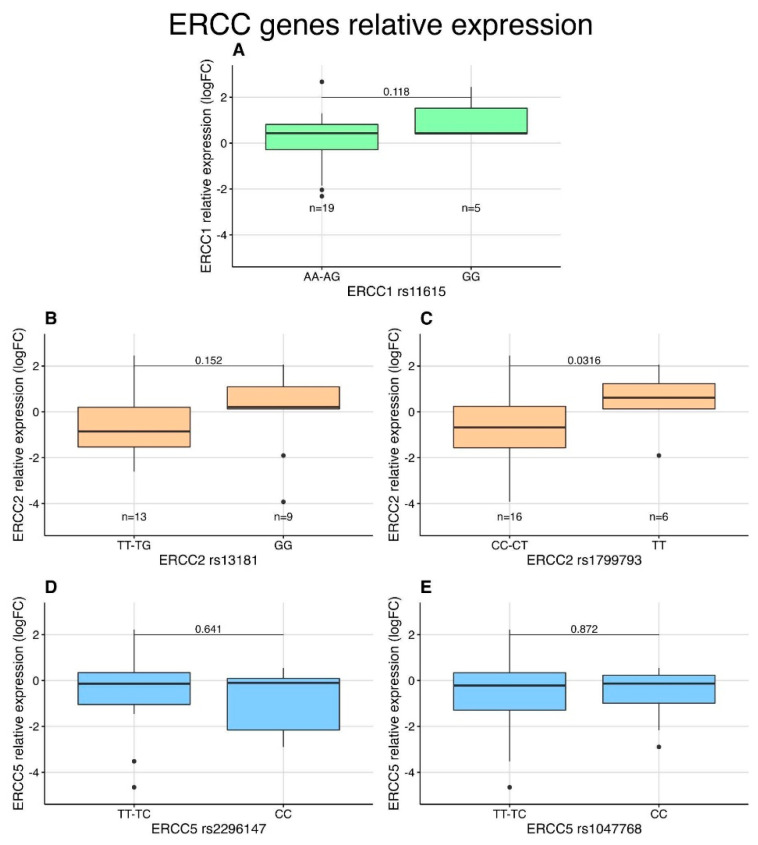
Correlation analysis between relative gene expression and genotype in 5 different SNPs (**A**) ERCC1_rs11615, (**B**) ERCC2_rs13181, (**C**) ERCC2_rs1799793, (**D**) ERCC5_rs2296147, (**E**) ERCC_rs1047768. The relative gene expression is presented as log2 Fold Change. In every plot, the samples were separated according to the presence (left side) or absence (right side) of at least one wild-type allele. The significance value at the top of every plot was obtained via Welch’s *t*-test. FC, fold change.

**Figure 6 ijms-23-08360-f006:**
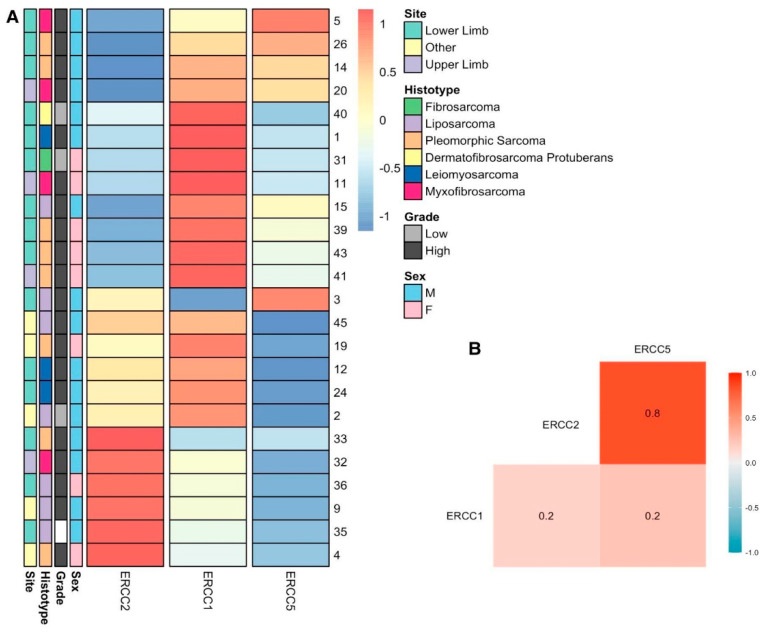
(**A**) Relative expression clusters of ERCC1, ERCC2 and ERCC5 per patient. The rows represent each sample and its respective ID, sex, histotype and grade. The columns represent the relative expression of ERCC1, ERCC2 and ERCC5 as log Fold Change values. (**B**) Correlation matrix of ERCC1, ERCC2 and ERCC5 relative gene expression with Pearson’s correlation similarity. M, male; F, female.

**Figure 7 ijms-23-08360-f007:**
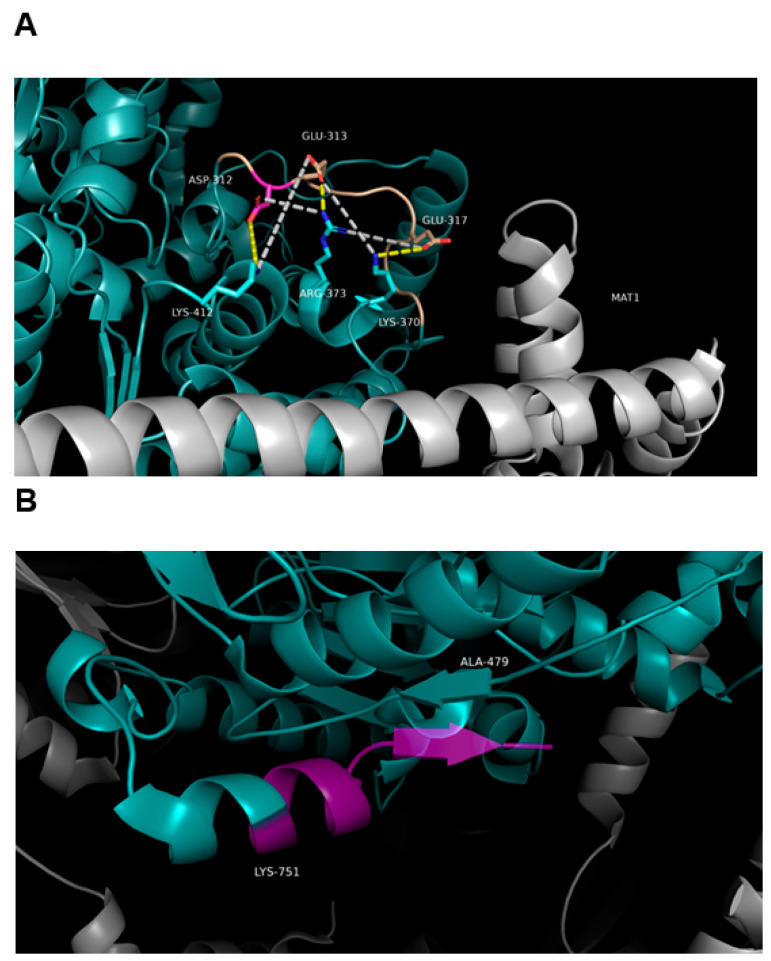
(**A**) Model of the ERCC2 electrostatic interaction network between the negative loop Pro311-Pro320 and positive charges; mutant Asp312Asn (position in magenta) destabilise the interaction and loop. (**B**) Loss of C-terminal α-helix and β-sheet (transparent segment in magenta) following mutation Lys751*.

**Figure 8 ijms-23-08360-f008:**
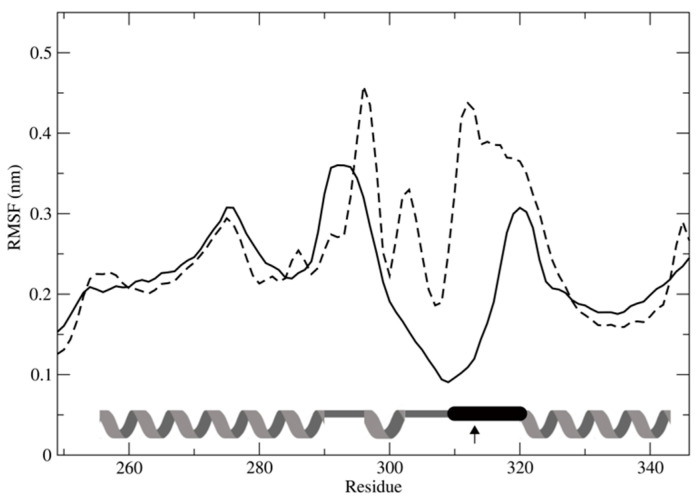
Protein local flexibility is represented with the root mean square fluctuation profile (RMSF, nm) for wild type (solid line) and Asp312Asn mutant (dashed line) as calculated from the molecular dynamics simulations. The secondary structure is reported at the bottom (helix for α-helices, line for loops). Loop segment Pro310-Pro320 (black thick line) shows flexibility lower than helices thanks to the extended network of electrostatic interactions. Point mutation Asp312Asn (indicated by arrow) causes electrostatic network disruption and segment flexibility to increase up to 4-fold. The RMSF of the remaining structure is not affected, suggesting only local conformational change occurs without hampering the overall folding.

**Figure 9 ijms-23-08360-f009:**
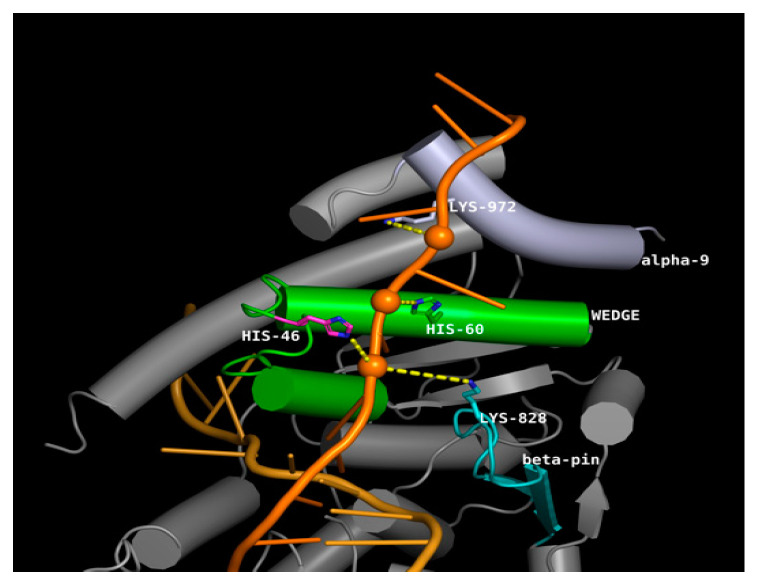
Model of activated ERCC5 using the DNA in the T4 RNase H crystal structure (PDB 2IHN). Close-up views of the gateway and the canyon from the model shown in panel B. Residues mutated in silico are colored in magenta. The undamaged ssDNA fits in the crevice formed by the hydrophobic wedge (green), the β-pin motif (cyan) and α-helix 9 (light blue). The crevice accommodates the phosphate backbone and is stabilised by charge interactions with bottom Lys828, His80 and Lys972; the MD simulation also showed wedge flexible loop to participate in binding by His46 driven closure onto the phosphate backbone. The His46Gln mutant largely lost such interaction.

**Table 1 ijms-23-08360-t001:** Clinicopathological features of the patients enrolled with STS.

Clinicopathological Features	Cohort (N = 32)
Gender	Male = 62.5%
	Female = 37.5%
Age at surgery (years)	Mean = 68.4
	Max = 93
	Min = 39
BMI (kg/m^2^)	Mean = 25.5
	Max = 44.3
	Min = 19.2
Surgical margins	Large = 90.6%
	Marginal = 9.4%
Primary/Relapse	Primary = 81%
	Relapse = 19%
Grade	High = 78.0%
	Low = 22.0%
Hystotype	Liposarcoma = 28.1%
	Pleomorphic sarcoma = 37.5%
	Leiomyosarcoma = 9.4%
	Myxofibrosarcoma = 12.5%
	Fibrosarcoma = 3.1%
	Fibromyxoid sarcoma = 3.1%
	Dermatofibrosarcoma protuberans = 3.1%
	Undifferentiated pleomorphic cell sarcoma = 3.1%
Anatomical site	Thigh = 62.5%
	Foot = 3.1%
	Gluteal-sacral region = 6.25%
	Scapular region = 3.1%
	Deltoid region = 3.1%
	Abdominal region = 3.1%
	Pectoral region = 6.25%
	Upper limb = 12.5%

Note. BMI, body mass index.

**Table 2 ijms-23-08360-t002:** Sequences of qPCR primers.

Gene Name	Primer Forward	Primer Reverse	Lenght (bp)
ERCC1	5′ CTCAAGGAGCTGGCTAAGATGT 3′	5′ CATAGGCCTTGTAGGTCTCCAG 3′	90
ERCC2	5′ CTGGAGGTGACCAAACTCATCTA 3′	5′ CCTGCTTCTCATAGAAGTTGAGC 3′	100
ERCC5	5′ CAGACACAGCTCCGAATTGA 3′	5′ TTCTGGGTTTTTCGTTTTGC 3′	209
18S	5′ AACCCGTTGAACCCCAT 3′	5′ CCATCCAATCGGTAGTAGCG 3′	151

Note: bp, base pair.

**Table 3 ijms-23-08360-t003:** Allelic and genotypic frequencies of SNPs in DNA extracted from buccal swabs.

Gene		ercc1	ercc2		ercc5	
SNP ID		rs11615	rs13181	rs1799793	rs1047768	rs2296147
**Assay ID**		C_2532959_20	C_3145033_10	C_3145050_10	C_1891769_20	C_1891782_20
**Chromosome Location**		chr19:45420395	chr19:45351661	chr19:45364001	chr13:102852167	chr13:102846025
**Variant**		A>G	T>G	C>T	T>C	T>C
**SNP Type**		Synonymous variant	Stop gained variant	Missense variant	Missense variant	Upstream variant
**Genotype**	**Homozygous Reference**	AA = 6 (19%)	TT = 6 (29%)	CC = 6 (19%)	TT = 6 (19%)	TT = 10 (32%)
	**Heterozygotes**	AG = 18 (58%)	TG = 14 (45%)	CT = 17 (55%)	TC = 14 (45%)	TC = 15 (49%)
	**Alternative Homozygotes**	GG = 7 (23%)	GG = 10 (36%)	TT = 7 (26%)	CC = 11 (36%)	CC = 6 (19%)
**STS cohort alleles frequencies**	**Reference**	48%	42%	47%	42%	56%
	**Alternative**	52%	58%	53%	58%	44%
**General population alleles frequencies ***	**Reference**	59%	64%	68%	42%	54%
	**Alternative**	41%	36%	32%	58%	46%
**χ² Test**	**χ²**	2.43	9.72	9.02	0.00	0.08
	** *p* **	0.1190	0.0018	0.0027	1.0000	0.7762

* https://www.ncbi.nlm.nih.gov/snp; https://www.snpedia.com; https://www.pharmgkb.org, accessed on 2 March 2022; STS, soft tissue sarcoma; chr, chromosome.

**Table 4 ijms-23-08360-t004:** Genotypic frequency of SNPs in the somatic line.

Gene	SNP ID	Homozygotes Reference (%)	Heterozygotes (%)	Homozygotes Alternative (%)
ERCC1	rs11615	100	0	0
ERCC5	rs1047768	80	20	0
ERCC5	rs2296147	87.5	12.5	0
ERCC2	rs13181	100	0	0
ERCC2	rs1799793	100	0	0

## Data Availability

Data are available upon request from the corresponding author.

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
