# Peer review of "Alteration of the Nucleotide Excision Repair (NER) Pathway in Soft Tissue Sarcoma"

_ijms, 2022, doi:10.3390/ijms23158360_

Round 1

Reviewer 1 Report

Review report

Alteration of nucleotide excision repair (NER) pathway in soft tissue sarcoma, by Adriano Pasqui et al.

Summary: Aim of the study is to present a severe deregulation of NER pathway in patients with soft tissue sarcoma,  and describe for the first time the effect of SNP rs1047768 in the ERCC5 structure suggesting a role in modulating 31 ssDNA binding.

Comments:

The manuscript is clear, relevant for the field and presented in a well-structured manner. References are pertinent. The manuscript is scientifically sound and the experimental design is applicable. The figures/tables/images/schemes are relevant, they properly show the data in most cases; data is interpreted clearly in most cases and consistently throughout the manuscript. Conclusions are consistent with the evidence and arguments presented.

Specific comments  

20-32.

The abstract does not follow the suggested abstract structure of IJMS. The background section makes up to 80% of the abstract. Please reconsider: Background, Results, Methods, Conclusion section according to the IJMS template.

32.

ssDNA – please clarify the abbreviation.

159-170

Results -Clinicopathological features of OS patients

The percentages have to be synchronized with Table 1.  (20 patients-  tumor localized in the thigh; histotype liposarcoma, relapsing, BMI)

99. Materials and methods

Interventionary studies involving humans require ethical approval. Please list the authority that provided approval and the corresponding ethical approval code.

194-196.

„For both SNPs in the ERCC2 194 gene this difference was statistically significant (Table #) (rs13181 X2=8,86 p=0,0029; 195 rs1799793 X2=8,21 p=0,0042).”

Please add Table number.

202.

Table 3. Allelic and genotypic frequencies of SNPs in DNA extracted from buccal swabs.

There is an unusual sign in coloumn rs1047768, last line. Please remove it.

205.

Figure 2. Presence/absence of the ERCC variants in the STS cohort.

The patient ID-s are not in due course, moreover, there are only 27 patients ID-s represented, not 32.  That may be the cause, the number of patients in the results section (line 197-201) and in Figure 2. are not similar. Please check and extend the figure or make the results in the text more clear and informative.

221.

All 32 patients were genotyped for somatic mutation? If not, how many and why are any missing?

236.

„ 24 tissue samples from the 32 patients in our cohort.”

8 tissue sample are missing, please clarify.

265.

Is there any information in the literature regarding association of female sex and higher expression levels of ERCC1?

What are the limitations ogf the study?

According to IJMS template, please add

1.     Author Contributions:

2.       Institutional Review Board Statement:  “The study was conducted in accordance with the Declaration of Helsinki, and approved by the Institutional Review Board (or Ethics Committee) of NAME OF INSTITUTE (protocol code XXX and date of approval).” for studies involving humans.

3.     Informed Consent Statement: Any research article describing a study involving humans should contain this statement. Please add “Informed consent was obtained from all subjects involved in the study.”

4.       Data Availability Statement

5.       Conflicts of Interest

Author Response

Reviewer #1:

The abstract does not follow the suggested abstract structure of IJMS. The background section makes up to 80% of the abstract. Please reconsider: Background, Results, Methods, Conclusion section according to the IJMS template.

In the revised version of the manuscript we have revised the abstract according to the suggestion of the reviewer.

ssDNA – please clarify the abbreviation.

In the revised version of the manuscript we have edited accordingly reviewer suggestion.

Results -Clinicopathological features of OS patients. The percentages have to be synchronized with Table 1.  (20 patients-  tumor localized in the thigh; histotype liposarcoma, relapsing, BMI).

According to reviewer suggestion we have revised Table 1 in the new version of the manuscript.

Materials and methods interventionary studies involving humans require ethical approval. Please list the authority that provided approval and the corresponding ethical approval code.

The informations have been added in the revised version of the manuscript.   

For both SNPs in the ERCC2 194 gene this difference was statistically significant (Table #) (rs13181 X2=8,86 p=0,0029; 195 rs1799793 X2=8,21 p=0,0042).” Please add Table number.

We have revised in the new version of the manuscript according to reviewer suggestion.

Table 3. Allelic and genotypic frequencies of SNPs in DNA extracted from buccal swabs. There is an unusual sign in column rs1047768, last line. Please remove it.

We have revised in the new version of the manuscript according to reviewer suggestion.

GENOTIPO

ALLELI

X2

P

ERCC1

rs11615

AA=6 19%

AG=18 58%

GG=7 23%

A=48 %

G=52 %

2.43

0.1190

ERCC2 (1)

rs13181

TT=6 19 %

TG=14 45 %

GG=11 36 %

T= 42 %

G= 58 %

9.72

0.0018

ERCC2 (2)

rs1799793

CC=6 19 %

CT=17 55 %

TT=8 26 %

C= 47 %

T= 53 %

9.02

0.0027

ERCC5 (2)

rs1047768

TT=6 19 %

TC=14 45 %

CC=11 36 %

T= 42 %

C= 58 %

0.00

1

ERCC5 (1)

rs2296147

TT=10 32 %

TC=15 49 %

CC=6 19 %

T= 56 %

C= 44 %

0.08

0.7762

Figure 2. Presence/absence of the ERCC variants in the STS cohort. The patient ID-s are not in due course, moreover, there are only 27 patients ID-s represented, not 32.  That may be the cause, the number of patients in the results section (line 197-201) and in Figure 2. are not similar. Please check and extend the figure or make the results in the text more clear and informative.

We apologize for inconvenience, we have properly revised in the new version of the manuscript

All 32 patients were genotyped for somatic mutation? If not, how many and why are any missing?

We have detailed in the revised version of the manuscript

24 tissue samples from the 32 patients in our cohort.” 8 tissue sample are missing, please clarify.

 For some patients tumor tissue was not available.

Is there any information in the literature regarding association of female sex and higher expression levels of ERCC1? 

To the best of our knowledge no data are available in literature regarding  association of gender and expression levels of ERCC1.

What are the limitations of the study? 

First of all, our study is a monocentric study and additionally includes a limited number of patients, so we need to validate collected data in a multicentric independent study.

According to IJMS template, please add.

We have revised in the new version of the manuscript according to reviewer suggestion and to IJMS template.

Reviewer 2 Report

In this manuscript, the authors reported their study on Alteration of nucleotide excision repair (NER) pathway in soft tissue sarcoma from 30(or 32) patients. The authors did quite a few analysis , such as sequencing,  gene expression and computational analysis. Especially, they found that  SNP rs1047768 in the ERCC5 structure might play an important role. Overall this is an interesting study. But the manuscript has some limitations and need to be addressed before publishing.

(1) Figure and Tables: some of the abbreviations need to be spell out and the numbers should be corrected . For example "81,2%"->"81.2%"

(2) Quite a few studies have been published on soft tissue sarcoma. Can you compare your study with published results? Maybe you can do similar analysis on those datasets.

(3) Do you have the analysis stratified by genotypes for outcomes(such as relapsing )? The concern I have is that the sample size is too small. Again, you can find more samples from the pubic datasets.

Author Response

Reviewer #2:

Figure and Tables: some of the abbreviations need to be spell out and the numbers should be corrected . For example "81,2%"->"81.2%"

We have revised in the new version of the manuscript according to reviewer suggestion.

Quite a few studies have been published on soft tissue sarcoma. Can you compare your study with published results? Maybe you can do similar analysis on those datasets.

To the best of our knowledge, few studies regarding NER pathway on soft tissue sarcoma have been published and with limited sample size. As of today we do not have access to this kind of data.

Do you have the analysis stratified by genotypes for outcomes (such as relapsing )? The concern I have is that the sample size is too small. Again, you can find more samples from the public datasets.

Our study (ie RESEARCH trial) is still enrolling patients, and data reported in this manuscript is a pilot analysis. We completely agree with the reviewer on the insufficient number of patients and we need to validate collected data in a multicentric independent study.

Round 2

Reviewer 2 Report

Again, Tables: some of the abbreviations need to be spell out and the numbers should be corrected . I don't feel that the authors considered very seriously on that.

Quite a few studies have been published on soft tissue sarcoma. Can you compare your study with published results? Maybe you can do similar analysis on those datasets. Again, I would suggest that the authors can compare their results with https://pubmed.ncbi.nlm.nih.gov/29100075/ . If no individual level data is available, you can use this interface to search by gene names https://www.cbioportal.org/study/summary?id=sarc_tcga_pub
